# The importance of multiregional accounting for corporate carbon emissions

Steven J. Davis [1,2] ✉, Andrew Dumit[2], Mo Li [2], Yohanna Maldonado[2], Michael Steffen[2], Martha Stevenson[3], Tatiana Boldyreva[4] & Sangwon Suh [2,5] ✉

Corporations routinely use environmentally-extended input-output models to estimate and report greenhouse gas emissions upstream in their supply chains. However, the most widely used models assume that supply chains and emissions intensities of industries match those of a single region—usually the U.S. or the U.K. Here, we use a high-resolution multiregional input-output model to demonstrate the scale and pattern of emissions that may be missed by single-region models. We find that the upstream emissions of the companies reporting to CDP are together greater by 2.0 gigatons of $CO_2$-equivalent emissions (~10%) when estimated by a multiregional model instead of a U.S.-based single-region model, with the largest differences in manufacturing sectors of moderate emissions intensity. Widespread adoption of multiregional models could thus improve the accuracy of corporate emissions inventories and help prioritize primary data collection and emissions reduction efforts, often by shifting focus to energy- and emissions-intensive sectors of industrializing nations.

Corporations began voluntarily estimating greenhouse gas (GHG) emissions related to their operations and business in the 1990s[1], supported since 2001 by the GHG Protocol Corporate Standard (a joint effort of the World Resources Institute and the World Business Council for Sustainable Development)[2]. But interest in corporate-level GHG accounting has surged in recent years, as companies increasingly prioritized sustainability and new regulations require emissions disclosures[3–5]. At the same time, companies are also increasingly making commitments to drastically reduce GHG emissions related to their business in support of international climate targets[6,7].

Corporate-level emissions inventories, or "footprints", commonly separate emissions related to a company's own activities (scope 1), emissions related to electricity or heat purchased and consumed by the company (scope 2), and emissions upstream or downstream in the value chain of the company's products or services (scope 3)[2]. Whereas companies typically have good records of their own GHG-emitting activities and purchases of electricity (scopes 1 and 2), they often lack detailed data about the emissions related to goods and services they

purchase (scope 3), which in many cases dominate the total corporate footprint, and thus pose substantial challenges to measure and manage[8–10]. Although such lack of data may contribute to companies underreporting their scope 3 emissions[7,11–14], it is also the main reason that companies routinely estimate their scope 3.1 emissions using secondary data, such as sector-specific "emissions intensity" (i.e., the average emissions per unit of monetary value of products or services produced by that industry sector)[15]. Although primary data specific to companies' supply chains are preferable when and where it can be obtained[2,16], industry average emissions intensities based on secondary data are instrumental to companies prioritizing such data-gathering and subsequent decarbonization efforts.

However, the fidelity of such sector-specific emissions intensities depends on the data and methods on which they are based, including whether and how finely industry sectors and source regions are differentiated[17]. Of the 624 companies that specified the source of their scope 3.1 emissions in corporate carbon footprints disclosed to CDP in 2023[18], 75% obtained such sector-specific emissions factors from a

[1]Department of Earth System Science, Stanford Doerr School of Sustainability, Stanford University, Stanford, CA, USA. [2]Watershed, San Francisco, CA, USA. [3]World Wildlife Fund, Washington, DC, USA. [4]CDP, London, UK. [5]Bren School of Environmental Management, University of California, Santa Barbara, Santa Barbara, CA, USA. ✉e-mail: sjdavis@stanford.edu; sangwon@watershedclimate.com

single-region environmentally-extended input-output (EEIO) model, such as the USEEIO modeled developed by the Environmental Protection Agency (EPA). The USEEIO model is based on U.S. economic and GHG emissions data, and the emissions factors produced reflect the activities and interactions of 411 different industry sectors. Although the USEEIO model's sectoral resolution (411 sectors) is quite high relative to other EEIO models in common use by governments and academics, it does not distinguish products and services produced in the U.S. from those produced in other countries or regions of the world. That is, USEEIO models business activities in the U.S. as if the U.S. were a closed economy, with no imports or exports (known as the "domestic technology assumption"). Yet it has long been recognized that the emissions intensity of aggregated imports to the U.S., Japan, and European countries is substantially higher than that of goods and services produced domestically[19–22], and researchers have developed consumption-based inventories for nations and multiregional models for businesses that account for the effects of international trade[23–25].

Although it is not entirely clear why single-region EEIO models remain so prevalent in practice despite the existence of multiregional options, a number of factors may be contributing, including the lack of minimum quality requirements under prevailing voluntary standards, and because single-region EEIO models tend to be sourced in public or government agencies and free of charge (e.g., the U.S. EPA and U.K. Department for Environment, Food & Rural Affairs), whereas multiregional models have been developed by independent academic or research institutions and are sometimes only available for commercial purposes under a paid license. Additionally, there is an inherent conservatism and inertia of accounting methods because of how difficult it is for corporates to change data sources and undertake the process of rebaselining and communicating the change (which often corresponds to an increase in scope 3 emissions) to stakeholders. Here, though, we demonstrate and quantify the large effects of such multiregional resolution on individual companies' emissions inventories, which may substantially alter the priorities of those companies' data-gathering and emissions reduction efforts. In the aggregate, such changes could have massive implications for corporate-led climate mitigation, shifting focus to energy- and emissions-intensive manufacturing in China, India, and other industrializing countries.

Details of our analytic approach are included in the Methods. In summary, we use multiregional and single-region versions of the Comprehensive Environmental Data Archive (CEDA) EEIO model[26], which was first published in year 2000 and has been regularly updated since, to evaluate differences in sector-specific emissions factors, and then assess the aggregate effect on both the emissions inventories and reduction priorities of various types of companies. Based on publicly available input-output tables and macroeconomic statistics, the CEDA model we use here maintains similar sectoral resolution to the USEEIO model (400 sectors), but adds international trade among 65 countries that collectively represent >90% of world GDP, and a "rest of world" region (n.b. a condensed version of the full 148 country model). The emissions intensities estimated by the multiregional model are much more consistent with published country- and industry-specific values than a single-region (U.S.) model. For example, Supplementary Fig. 1 shows the range of emissions intensities for cement, steel, and electricity as estimated by both single-region and multiregional models as well as reported values from other sources. The python programming language was used for all computation. Note that an open data version of the CEDA model (CEDA 2024) has recently been made free and available, which may reduce barriers to adoption of multiregional models.

## Results

Among the ten industries that most commonly disclose their emissions to the CDP[18], we find substantial differences in average upstream emissions related to goods and services they purchase (i.e., scope 3.1) between the single-region (U.S.) and multiregional models, with multiregional inventories consistently larger (Fig. 1). In each case, the observed differences represent the scale by which a single-region model based on the U.S. economy may underestimate companies' scope 3.1 emissions. The largest differences are in manufacturing of structural products (+71.3%), construction machinery (+69.7%), fabricated metal (+50.6%), and electronic components (+39.3%), with more modest differences in chemical product and plastic manufacturing (+2.2% and 1.2%, respectively), business support services (+10.1%), financial investments (+6.8%), truck transportation (+10.5%), and software publishing (13.3%). These differences indicate that imports to the U.S. are consistently more emissions-intensive than similar goods produced domestically. Indeed, where differences are large we find that much of the additional emissions are often related to energy-intensive sectors in regions with carbon-intensive energy systems, such as iron and steel or resin production in China and Russia.

For example, examining the key upstream sectors of the same commonly-disclosing industries, we find that emissions intensities vary enormously across regions, with standard deviations in each case >180 gCO$_2$e/$ (Fig. 2a). In some cases, such as electronics manufacturing, the single-region (U.S.) model suggests very low emissions intensity (vertical black lines in Fig. 2a) relative to the multiregional distribution (colored density plots). This may be because large U.S. companies in this category, such as Apple Inc. are primarily engaged in design and branding and less of the energy-intensive manufacturing than electronics firms in other regions. However, the U.S. value is not consistently higher or lower than the central tendency of these regional distributions (vertical lines in Fig. 2a). Sectoral resolution (i.e., the level of granularity in sector classification) also remains important: Fig. 2b shows the similarly wide distributions of emissions intensity of industry sectors within broader industry categories (gray density plots with 1σ > 150 g CO$_2$e/$) in comparison to the multiregional median of the specific sector of interest (colored vertical lines).

Comprehensively assessing differences in sectors' emissions intensities as estimated by single-region (U.S.) and multiregional models, we find that the multiregional model intensities are generally higher, but there are 29 sectors (7%) in which the single-region model intensities are higher, including plastic bottle manufacturing, pesticide and agricultural chemical manufacturing, and fats and oils refining and blending; Fig. 3a). In particular, the emissions intensities of manufacturing sectors are often >20% higher according to the multiregional model (light orange circles in Fig. 3a, b). Relatedly, the greatest differences in the multiregional results are concentrated in sectors with midrange emissions intensities (between 0.4 and 0.8 kgCO$_2$e/$; Fig. 3c), which suggests that energy efficiency and sources of energy used for lighter manufacturing are more variable than those of the most emissions-intensive sectors, such as cement and steel.

Multiregional and single-region model estimates of corporate footprints in many cases differ with respect to the location and magnitude of hotspots in upstream emissions. Figure 3d highlights how the ranking of the emissions sources across all 400 industry sectors changes when using the multiregional model: darker shaded cells indicate cases in which the multiregional model ranks the emissions of a contributing (row) sector higher than the single-region model for a sector of interest (column). In particular (and consistent with the sectoral comparison in Fig. 3a), the multiregional results reveal emissions hotspots in the manufacturing sector (as well as the sectors related to equipment repair which often also entail manufactured parts).

As with sectors, the differences in emissions estimated by multiregional and single-region models are also unevenly distributed by region. Colors on the map in Fig. 4 indicate the scale

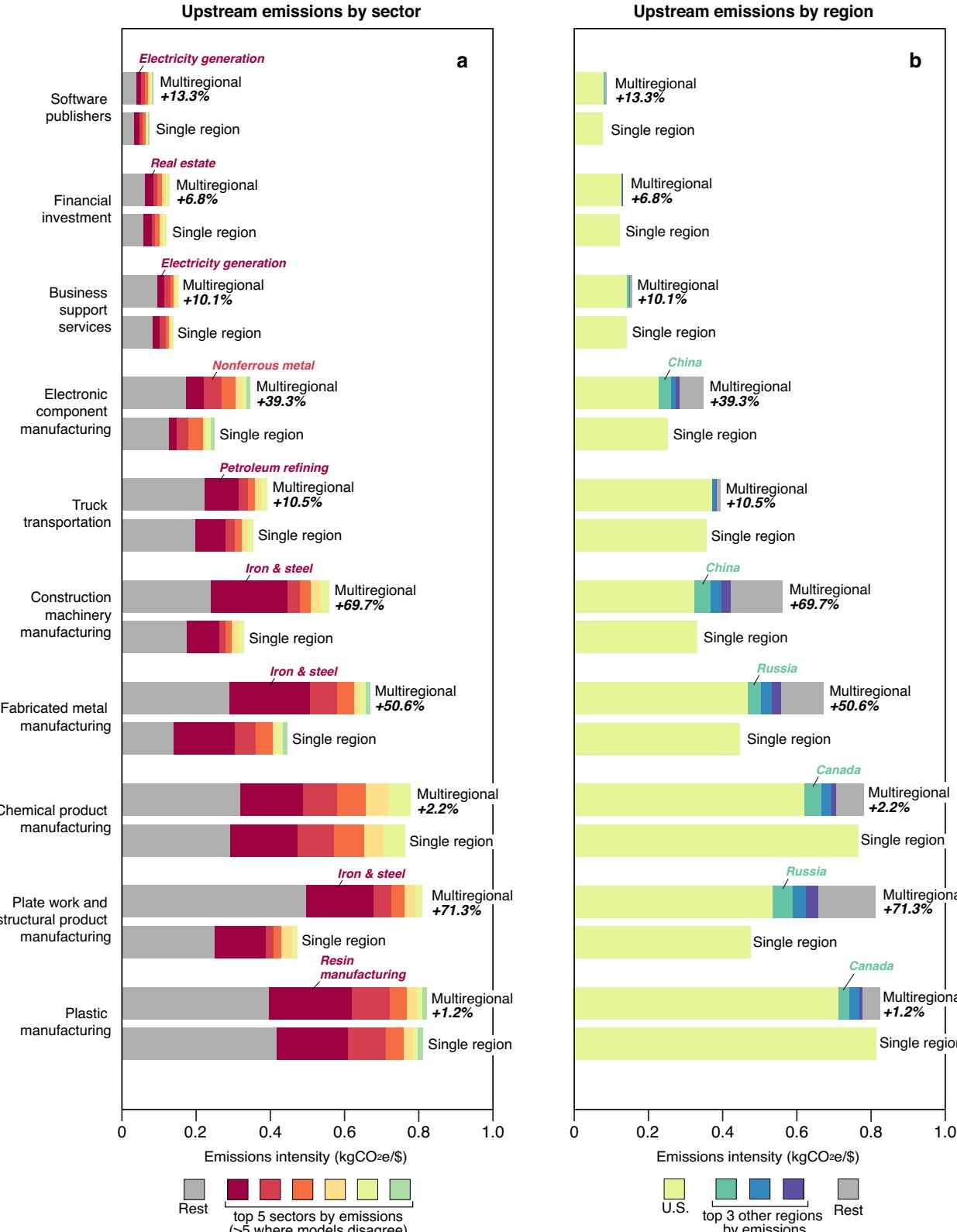

**Fig. 1 | Sector-specific differences in upstream emissions due to multiregional resolution.** Paired bars show differences in the industry average upstream emissions related to purchased goods (i.e., scope 3.1) per dollar of products or services produced among the top ten industry sectors of companies reporting their emissions to CDP 2021–2023 calculated by single-region (U.S.-specific) and multiregional input-output models. The differences are further decomposed as they relate to specific sectors (**a**) and regions (**b**). In all these cases, the single-region model underestimates upstream emissions.

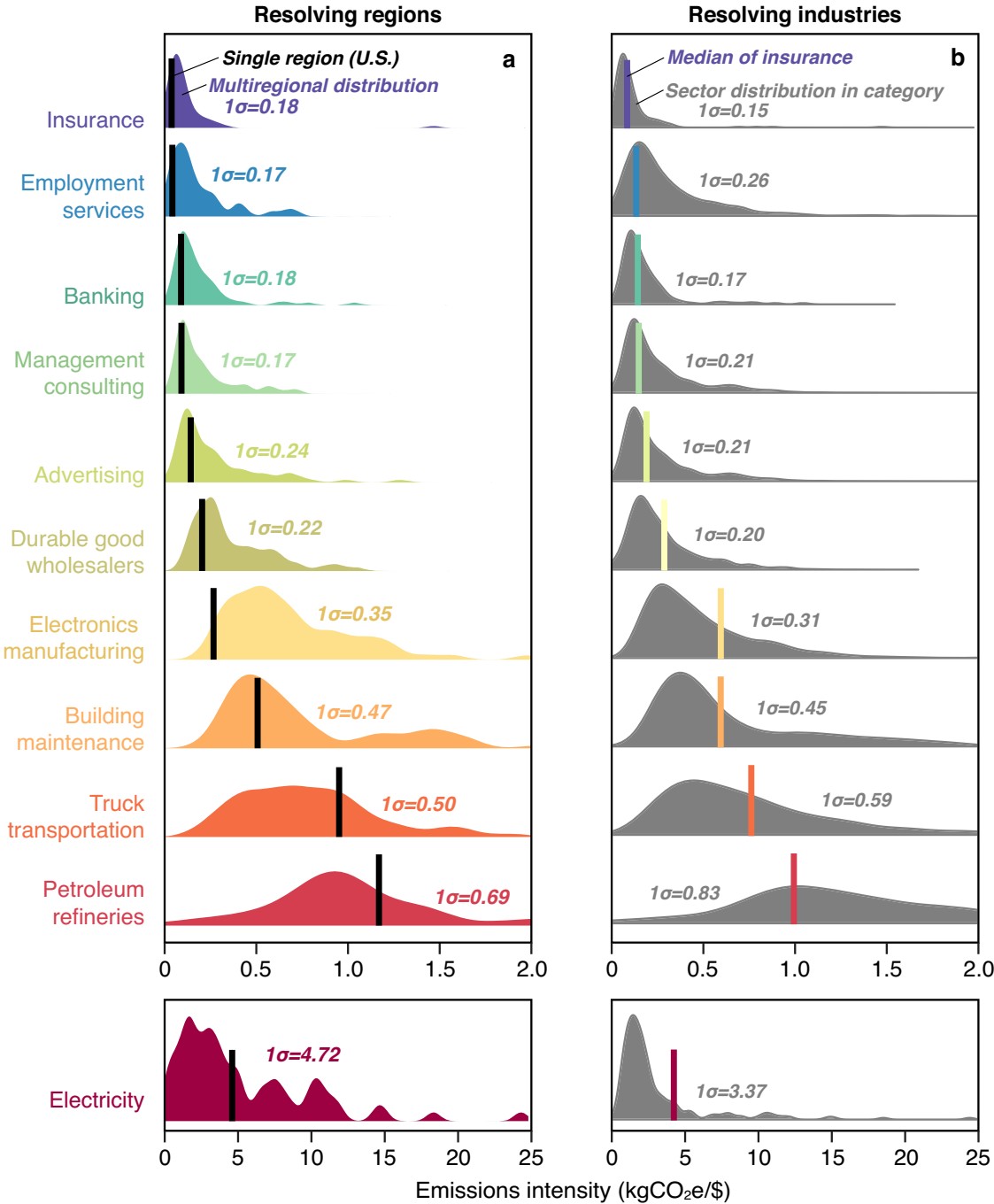

**Fig. 2 | Distributions of emissions intensity in key supplier sectors.** Across regions, there is wide variation in the average emissions intensity (emissions per dollar of products or services produced) of key supplier sectors (colored probability density plots in (**a**), such that the emissions intensity from a single-region (U.S.-based) model (black lines in (**a**) may substantially over- or underestimate the reality. Similarly large variation in the average emissions intensity of specific sectors within broader industry categories (gray density plots in (**b**), such that even multiregional models with fewer (more aggregate) industry sectors may also over- or underestimate the emission intensity of a specific sector (colored lines in **b**) by a similar margin.

of differences in country-level emissions when using the multiregional rather than single-region (U.S.) model to estimate upstream emissions of all the companies that report their emissions to CDP (a total difference of 2.0 GtCO$_2$e globally, which is 9.7% of total emissions reported to CDP by those companies in 2023). The dark red of China thus reflects both that country's outsized role in international supply chains as well as often greater emissions per unit of Chinese production: in the aggregate, CDP-reporting companies using a U.S.-based single-region may miss >900 MtCO$_2$e of related emissions occurring in China

(Fig. 4). On the other hand, emissions in other countries may be overestimated by single-region models for example where low-carbon energy sources are a larger share of energy used than in the U.S., as with nuclear in France or hydroelectricity and bioenergy in some South American countries (blue shading in Fig. 4). The arrows shown in Fig. 4 highlight the largest international transfers of emissions embodied in CDP-reporting companies' upstream supply chains according to the multiregional model. The prevalence of arrows out of China thus reflects the importance of Chinese production in these companies' supply chains

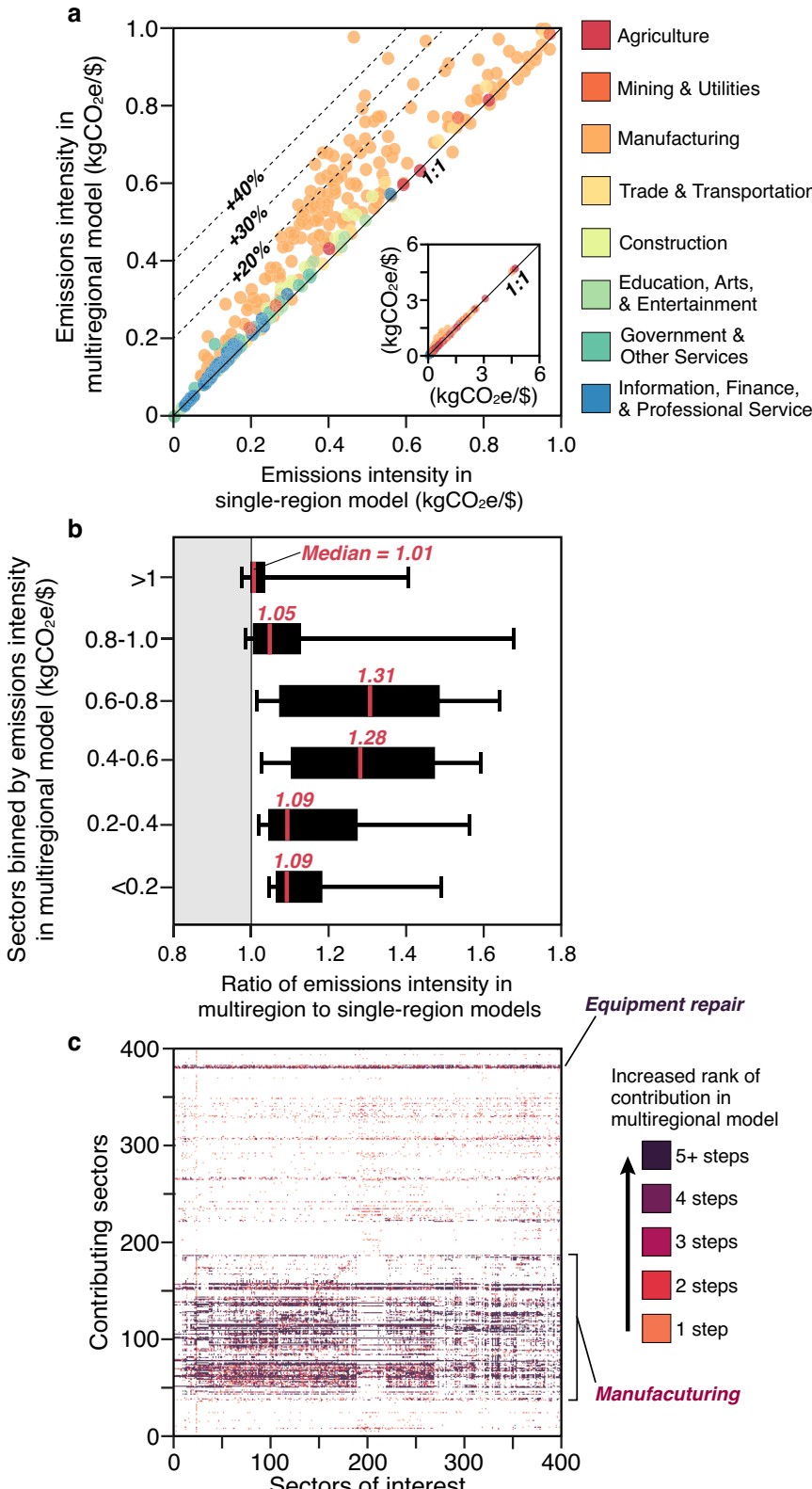

**Fig. 3 | Comprehensive comparison of sector-level differences between single-region (U.S.-based) and multiregional models.** Across all 400 industry sectors, the emission intensities (emissions per dollar of products or services produced) estimated by the multiregional model are generally greater than those estimated by the single-region model (points above 1:1 line in (**a**), particularly among manufacturing sectors (orange points). Grouping sectors according to their emissions intensity as estimated by the multiregional model shows that the emission intensities from the multiregional are most different (roughly 30% greater) in sectors with emissions intensity of 0.4–0.8 kgCO₂e/$, and the rare cases in which the single-region model estimates greater emissions intensity than the multiregional model are mostly in sectors which very high emissions intensities (>0.8 kgCO₂e/$; **b**). Colors plotted in **c** indicate the magnitude of increases in the rank order of contributing (upstream) sectors (y-axis) as sources of emissions to different sectors of interest (x-axis) when using the multiregional rather than single-region model.

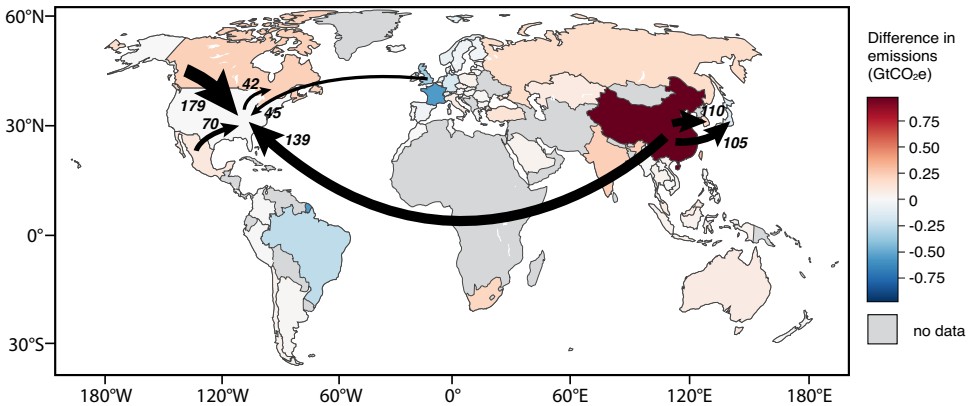

**Fig. 4 | Map of differences between single-region (U.S.-based) and multiregional EEIO models.** Shaded colors indicate country-level differences in emissions when estimating upstream emissions of CDP-reporting companies using the multi-regional model instead of a single-region (U.S.) model. In total, the multiregional model estimates 2.0 GtCO$_2$e more emissions worldwide than the single-region model, but international supply chains and higher emissions-intensities of production in China lead to much greater emissions in China (+973 MtCO$_2$e), and somewhat lower emissions in areas which rely more heavily on low-carbon sources of energy (e.g., France, Brazil, and the U.K.). Arrows highlight the largest international transfers of emissions embodied in these companies' upstream supply chains that are missed by a single-region model. Made with Natural Earth base map.

while arrows into the U.S. reflect the disproportionate size and proportion of U.S. companies among CDP-reporters.

## Discussion

Our analysis demonstrates that there may be substantial and consequential differences in corporate carbon accounts when resolving multiregional sources of upstream emissions. Such differences reflect the large regional variabilities in production technologies and energy sources not captured by single-region models. And although we have mostly focused on the inaccuracies that may result due to those existing regional differences, distinct regional trends may also lead to erroneous projections of future emissions if based on single-region models. These trends, and the reliability of relative year-to-year changes are arguably much more important for reduction planning and monitoring than the accuracy of a company's carbon footprint in a single year. Similarly, the representativeness of emissions estimated for a given company depend not only on the methods being used, but that the full scope and type of a company's activities are being assessed. Sectoral and regional resolution, as well as up-to-date data, are important factors, but attention must also be paid to boundary-setting and the quality of activity data being used (i.e. expenditures).

In the common situation of U.S. companies that have been using the U.S. EEIO model to estimate upstream emissions, switching to a multiregional model may lead to increases in their corporate footprint. Where such multiregional resolution is still not required by standards or regulations, many companies may therefore continue to use single-region models, which may ultimately lead to under-estimation of their footprints and misallocation of mitigation efforts. For example, insofar as U.S. climate policies are more ambitious than those in other regions, corporations may not only underestimate their footprints but neglect otherwise cost-effective reduction opportunities that might be realized by engagement with international suppliers: Reflecting regional differences and changes in the emissions intensity of electricity over time, our multi-regional model shows that the upstream emissions per dollar of revenue of selected Indian industries trends up 2010–2020, while the same industries' emissions intensities trend down when a single-region U.S.-based model is used (Supplementary Fig. 2a). Similarly, the top three hotspots of upstream emissions in either case are substantially different (Supplementary Fig. 2b). Aggregated across voluntary corporate efforts, such missed opportunities could conceivably undermine the efficacy of corporate climate action. For example, assuming all the

companies reporting their emissions to the CDP in 2023 were to estimate their upstream emissions using one or the other type of model, the multiregional model captures 2.0 Gt CO$_2$e more emissions in the aggregate than the single region model (which is 9.7% of total emissions reported to CDP in 2023 and ~14% of global CO$_2$ emissions in the same year[27,28]—note that there may be multiple counting of scope 3 emissions by different companies within a value chain). In contrast, where governments plan to regulate emissions embodied in imports (e.g., the E.U.'s Carbon Border Adjustment Mechanism[29,30]), investors relying on emissions disclosures from a single-region model may substantially underestimate businesses' exposure. Future work might productively explore the underlying sources of regional differences in emissions to reveal, for example, the extent to which they can be explained by regional differences in energy sources as opposed to the type and efficiency of industrial processes.

Several limitations and caveats apply to our results. First and foremost, the emissions intensities derived from EEIO models represent industry averages. Regardless of whether the model is multiregional or single-region, primary activity data from a company's suppliers is generally preferred. However, footprint uncertainty may be minimized by hybrid approaches that use supplier-specific activity data to adjust industry averages within a multi-regional input-output model, especially if efforts to collect such activity data target upstream hotspots[31]. Secondly, the version of the CEDA model used in this study does not include emissions related to land-use change, which are a substantial and somewhat uncertain global emissions source that is extremely heterogenous across regions[32]. Third, as demonstrated in Fig. 1b, sectoral detail is approximately as important as regional resolution: using a multi-regional EEIO model with much more aggregated sectors (e.g., EXIOBASE[33]) risks increasing overall uncertainty. Fourth, companies might use multiregional resolution to seek out and switch to suppliers in less emissions-intensive regions rather than engaging to improve energy systems and processes in their existing supply chains. Although such a strategy might reduce these companies' upstream emissions footprint (as attributed to their current supply chains using existing emissions intensities), it could undermine the overall climate benefits of corporate climate action insofar as the suppliers in more emissions-intensive regions can find new corporate buyers not concerned with their GHG emissions (perhaps due to a lack of multiregional emissions accounting). This sort of leakage

**Table 1 | CEDA Data Sources**

| Data type | Source and Reference | Base year |
|---|---|---|
| Input-Output tables | U.S. Bureau of Economic Analysis (BEA)[49] | 2012, 2018 |
| | U.K. Office for National Statistics (ONS)[50] | 2018 |
| | National Bureau of Statistics of China (NBS)[51] | 2015 |
| | Bank of Korea[52] | 2018 |
| | Statistics of Japan[53] | 2018 |
| | Organization for Economic Co-operation and Development (OECD)[48] | 2018 |
| International trade statistics | U.N. Comtrade[54] | 2018 |
| | Organization for Economic Co-operation and Development (OECD)[48] | 2018 |
| GHG emissions inventories | U.S. Environmental Protection Agency (EPA)[55] | 2018 |
| | U.K. Office for National Statistics (ONS)[56] | 2018 |
| | National Bureau of Statistics of China (BNS)[51] | 2014 |
| | Greenhouse Gas Inventory & Research Center of Korea[57] | 2018 |
| | National Institute for Environmental Studies of Japan[58] | 2018 |
| | United Nations Framework Convention on Climate Change (UNFCCC)[59] | 2018 |

motivates consequential approaches that seek to evaluate the effects of an intervention relative to a counterfactual (e.g., the estimated difference in emissions when changing suppliers), which may complement attributional approaches by capturing system-wide impacts of corporate decisions[34], but have yet to be widely adopted by the standards used for corporate emissions accounting[35–37]. Fifth, spatial misspecification by single-region models represents only one of several contributors to uncertainty in current Scope 3 accounting practices. Although comprehensive, quantitative assessments of the relative contribution of each source remain scarce, the results presented here suggest that spatial misspecification can constitute a material share of total uncertainty in Scope 3 estimates. Continued and coordinated efforts are therefore warranted to reduce uncertainties across all principal sources. Moreover, input–output models represent only one of several methodological approaches for quantifying global supply-chain emissions, and the results presented here may not be directly reproducible in studies employing alternative methods or datasets. Finally, because the quality and availability of data on GHG emissions, economic structure, and international trade are geographically uneven and rapidly evolving, the CEDA model uses various methods and assumptions to fill gaps for those countries where such data are either not available or available at a lower resolution or quality than elsewhere (see Methods).

Nonetheless, our results show that large regional differences in the emissions intensities of industry sectors are important for the individual corporations which are increasingly estimating their own footprints. Not only do these difference affect the overall accuracy of a corporate footprint, they often substantially alter geography and rank order of upstream emissions sources, which may in turn lead to different priorities for both collection of supplier-specific activity data and emissions reduction efforts. In the aggregate, the shift in such priorities afforded by multiregional resolution could fundamentally alter the actionability and efficacy of corporate efforts by unblinding companies to upstream emissions that are amenable to mitigation. Insofar as standard-setting organizations like the GHG Protocol as well as regulators in government are interested in improving the accuracy, credibility, and usefulness of compliant corporate footprints, our findings emphasize the opportunity to require multiregional models rather than single-region models when estimating upstream scope 3 emissions.

## Methods

Our analysis and results are based on CEDA v5.0, a multiregional, environmentally extended input-output (EEIO) model. Such models theoretically avoid cutoff errors because their system boundaries encompass the entire global economy. CEDA was first released in 2000 and has been regularly updated since. To ensure consistency between the multiregional and single-region results, we create a single-region CEDA model for the United States (US) by endogenizing imports to domestic production, which is common practice among single-region EEIO models. This treatment of imports is equivalent to assuming that all goods and services worldwide are produced in the US with U.S. technologies.

The method and data sources used for the compilation of CEDA was published in 2005[26]. In the years since, the model has undergone regular updates and improvements, with the latest version (v5.0) encompassing 400 sectors across 148 countries and regions which are classified into three tiers (see Model Construction below). For simplicity, in the current study we focus on 65 countries that belong to the first and second tiers.

Sectors in the CEDA model are defined according to industry and commodity classifications of the US Bureau of Economic Analysis (BEA), which in turn follows the United Nations' standard accounting framework, the System of National Accounts (SNA)[38]. The same sector classification is used in all 148 countries and regions covered by CEDA.

### CEDA data sources

CEDA is constructed by using and reviewing over 100 data sources of three main types: (1) national input-output tables, (2) international trade statistics and (3) GHG emissions inventories. Table 1 summarizes the main sources and their corresponding base years.

In addition to these data sources, the CEDA model is validated against multiple, independent statistics and data sources including the European Union's Emissions Database for Global Atmospheric Research (EDGAR)[39], the Global Carbon Project database[27], and numerous statistics and reports[40–44].

CEDA compilation follows the key principle of identifying and using the best available data as well as using assumptions, models, and proxies in the event that suitable data is not available. Our methods are described in more detail in the following section.

### CEDA model construction

Here, we adopt common notations and matrix algebra used in EEIO models[33,45] and LCA[46,47]. The overall balancing equation for the flows of goods and services in monetary terms is shown as:

$$x = Ax + y \tag{1}$$

where $x$ is a vector of total commodity output by region, $y$ is a vector of final demand of commodity by region combination), $A$ shows the ratio of commodity input per unit of output, which is commonly referred to as technology coefficient[46,47]. The equation can be solved for $x$: $x = (I − A)^{-1}y$, where $(I − A)^{-1}$ is the Leontief inverse matrix, or $L$, and $I$ is an identity matrix. The Leontief inverse matrix captures the total (direct and indirect) input requirements from all sectors needed to produce one unit of final output in each sector, accounting for the complete supply chain effects throughout the economy. In a multiregional input-output model, $A$ is constructed in such a way that domestic intermediate economies are shown in the block diagonal matrices, e.g., $A_{1,1}$, while imports and exports are shown in off-diagonal matrices, e.g., $A_{1,r}$ (Eq. 2).

$$\begin{bmatrix} A_{1,1} & \cdots & A_{r,1} \\ \vdots & \ddots & \vdots \\ A_{1,r} & \cdots & A_{r,r} \end{bmatrix} \qquad (2)$$

Next, the emissions intensity matrix, $B$, is calculated as:

$$B = Ex^{-1} \qquad (3)$$

where $B$ is direct emission intensity of each sector in all regions, and $E$ is national total emissions from each production activity and region combination.

Finally, the direct and indirect emission intensity of each sector in all regions is represented by:

$$M = BL \qquad (4)$$

In practice, construction of CEDA is a more complicated approach than simple matrix multiplication because of different data granularity and availability for different countries included in CEDA. To solve this, we develop a Tiered Approach framework (Tiers 1, 2, and 3), described below.

We categorize Tier 1 countries as the U.S., the U.K., Japan, South Korea, and China. This categorization is based not only on their substantial roles in global trade and GHG emissions but also on the availability of the highest quality data sourced from each country's statistical reports on national input-output (IO) tables, energy flow, and GHG emission inventories. Tier 1 countries have relatively more granular and up-to-date IO tables and emissions data, which sets the foundation for CEDA's canonical sectoral classification.

Tier 2 countries encompass 60 countries featured in the OECD dataset, derived from excluding the 5 Tier 1 countries from a total of 65 OECD countries. National IO tables and GHG emissions data of Tier 2 countries are obtained from the OECD dataset where economic sectors are much more aggregated compared to those in Tier 1 countries[48]. Therefore, we develop a Structural Reflection technique to disaggregate national data of Tier 2 countries from OECD classification to CEDA's canonical 400 sector/product classification using the most closely related economy's highly detailed data. First, the country-level tables are disaggregated using Structural Reflection approach, and the resulting IO and GHG emissions data are cross-examined among themselves and across alternative data sources for manual adjustments. The resulting tables are used to calculate the GHG emissions embodied in each product by all 60 countries. These results are again cross-examined among themselves and across alternative data sources to identify outliers and anomalies.

Tier 3 countries do not have coherent IO statistics and are excluded from our analysis in this study.

## CDP Data

Of the 10,867 companies from 112 unique countries that reported their emissions to CDP in 2023, 4446 companies (40.9%) reported their scope 3.1 emissions, and 2800 (25.8% of the total) disclosed that they used a spend-based approach to estimate those upstream emissions. Among those 2800, 624 companies (i.e., 14% of those reporting scope 3.1) further specified the type of EEIO model used: 75% used a single-region EEIO model, and 25% a multiregional model. Here, we model emissions of all 5450 companies that reported their revenues to CDP in 2023. Using a single-region (U.S.) EEIO model, we estimate these companies' aggregate upstream emissions were 12.2 GtCO2 in 2022. If we instead use a multiregional model, we estimate their upstream emissions were 14.1 GtCO2—a difference of 2.0 Gt of emissions worldwide.

## Reporting summary

Further information on research design is available in the Nature Portfolio Reporting Summary linked to this article.

## Data availability

The raw inputs to the CEDA input-output models are publicly available and listed in Table 1. In turn, the resulting emissions factors from the CEDA 2024 model are published as open data here: https://openceda.org/solutions/ceda. CDP data are available under license, access can be obtained by via the website: https://www.cdp.net/en/data.

## Code availability

The code used to analyze CEDA and CDP data and generate plots for this manuscript is publicly available here: https://github.com/ProfFate/multiregional-acct/ and in a citable version on Zenodo with https://doi.org/10.5281/zenodo.17355134.

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

## Acknowledgements

The authors are grateful to Pankaj Bhatia, David Burns, Michael Macrae, and Wes Ingwersen for helpful discussions.

## Author contributions

S.J.D., Y.M., M.St., and S.S. conceived the analytical approach of the study. A.D. performed the input-output modeling. M.L., Y.M., and S.J.D generated the figures. S.J.D. led the writing with input from M.St., M.St., T.B., and S.S. All authors reviewed the paper.

## Competing interests

S.J.D., A.D., M.L., Y.M., M.Stef. and S.S. are employees of Watershed Technology, Inc., a company that licenses an enterprise version of the CEDA model and have an ownership stake in the form of shares and/or options in the company. The remaining authors have no competing interests.
