## [Transparent Peer Review file · Nature Communications]

The importance of multiregional accounting for corporate carbon emissions

Corresponding Author: Dr Steven Davis

Version 0:

Reviewer comments:

Reviewer #1

(Remarks to the Author)

This study addresses a methodological problem in corporate supply chain carbon accounting. While estimating Scope 3 emissions by using company-specific primary data yields the most accurate and specific estimates, such data is typically hard to obtain and therefore is often substituted with input-output data. I-O data sources such as the US-EEIO however rely on single-regional data which, this paper argues, results in undercounting of emissions. It demonstrates this by considering the use of multiregional models as a solution using a high-resolution multiregional input-output model (CEDA). The authors quantify the discrepancies in emissions estimates for upstream supply chains of companies, finding a substantial underestimation when using a single-region (US alone) model and analyze this discrepancy across sectors.

This paper, in my opinion, adds to the evidence of the importance of using multi-regional models. To this extent, this is an appreciable contribution.

The paper is also well written and easy to read though the explanations for Figures 2 and 3 can be more precise and need additional clarity. The analysis is competently executed, the methods clearly explained.

My major concern with the paper is that while this indeed adds to the body of evidence of the importance of multi-regional models, is this sufficiently novel in and of itself? There are several studies (e.g., see Li et al. 2020 Environ. Sci. Technol. DOI: 10.1021/acs.est.9b05245) in the literature arguing for and carrying out an analysis of carbon footprinting using MRIO models – the relative contribution of the present study must therefore be carefully explained and substantiated.

My assessment of the paper will be more favorable if the authors therefore do a more thorough and careful job positioning their study with the literature to provide a more complete picture of their original contributions and arguments to the reader.

Reviewer #2

(Remarks to the Author)

Find attached my comments in uploaded review file.

Reviewer #3

(Remarks to the Author)

GENERAL COMMENTS

Your manuscript is relevant to an interesting and policy-relevant issue, which is the uncertainty in and interpretation of corporate Scope 3 (category 1) emission estimates. The use of and reliance on EEIO emission factors (EFs) by companies is certainly important to understand. Unfortunately, I found your manuscript wanting with respect to both presenting a clear research question and then providing a scientific and/or policy-relevant discussion and answer.

Many of your findings seem to actually be about the differences in "spend-based" sectoral EFs between the two models you

examine. This is an interesting, although I assume not a novel, observation. A discussion of how EEIO models handle (or neglect) international trade flows is a good one. Exploration of the causes of uncertainties in these EFs would one research question to explore more, and could be done without any reference to voluntary corporate GHG inventory reporting.

Another research question would be the uncertainties and biases in voluntary corporate GHG inventory reporting that result in the use of different types of spend-based EFs and then the implications for those findings for one or more specific applications of corporate GHG reporting. It seems this is closer to what your manuscript is attempting to answer, but I did not find the discussion really explored any specific implications. The reader is then left wondering why does this matter. Is the finding that companies should shift to using a model like CEDA for all their scope 3.1 EFs? In all cases? What would be the implications of switching? You refer to how primary data is preferred but do not address the general uncertainties of using spend-based EFs from EEIO models in place of primary data.

In other words, your manuscript alludes to the findings having policy implications, but the reader is left struggling to clearly see them.

Overall, it would be useful if you would be clear and explicit regarding which specific use cases for corporate GHG reporting you are considering and referring to in your discussion. Is it just corporate net zero-type target setting and tracking? CBAM type policies? Security investor disclosure requirements/laws? Others?

SPECIFIC COMMENTS

Abstract, L40

It would be useful to have a sense of scale for the 2 Gt value (e.g., as a percentage of CDP reported totals or something).

Abstract, L46

The conclusion seems to be focused on accuracy (reduced uncertainty), but the manuscript spends little time addressing the issue of representativeness of estimates to a particular company or the sensitivity of estimates to changes in companies supply chain (time series trend uncertainties), which one could argue is the more important dimension of accuracy for many uses of corporate reporting.

L57

I believe the GHG Protocol is now referring to scope 2 as "purchased and consumed"

L68-69

We are not aware of any company that does not rely on industry average spend-based estimates for most or all of their scope 3.1. So it is more than "often instrumental".

L85-86

The phrasing here regarding how national inventories "need" to account for international trade effects is likely confusing or misleading. There are few consumption-based national GHG inventories and even fewer that have any policy relevance. So, it would be good to clarify that you are only referring to consumption-based inventories with this statement and that the point is irrelevant to the dominant production-based GHG inventory reporting that all countries do.

L103-114

The discussion of the differences in estimates across sectors is interesting, however, the discussion of why seems lacking. Even if it is necessarily speculative, it would be useful to elaborate here or in the discussion section below more on the analytical or physical causes for these differences.

L148-149

Again, some relative reference to the scale of this difference would be helpful.

L166-167

This finding is interesting, but no mention of the implications for emission trends is discussed. Further, are there boundary issues between the two models? Does one model implicitly create a wider value chain boundary for an "average company" than the other model? Do these differences reflect real regional differences or just structural differences in how each model draws supply chain boundaries for a company?

L170-172

Few if any GHG regulations in the USA or elsewhere are administered at the company-wide level versus the installation level, especially with regard to scope 3 emissions. So, it is not clear how or why your findings would have major implications for policy/regulation.

L179-181

In the case of CBAM, this is administered based on process-specific production data for the targeted major commodities. EEIO modeling is not generally used, nor are broader supply or value chain emission estimates for these commodities. So, it is not clear how or why the EEIO differences you find would be relevant to this policy.

L198-203

The point here is an argument for the use of consequential impact methods.

L207-210

This conclusion regarding scope 3 seems quite obvious for most companies that report to CDP, which is more biased towards larger multinationals. And probably for any company given international trade flows.

Version 1:

Reviewer comments:

Reviewer #1

(Remarks to the Author)

The authors have sufficiently addressed my comments.

I notice a few minor issues that the authors can correct or clarify.

1. There is a reference to a forthcoming CEDA 2024 dataset with a placeholder link. I am not sure if this was intentional or a typo, please correct this.

2. The discussion around attributional vs consequential approaches can be clearer, the authors can cite some relevant papers that get into this distinction more.

3. Finally, I also suggest the authors engage more directly with the supply chain management literature on supply chain emissions and Scope 3 emissions, particularly from top management journals. In my opinion, there is a bit of a disconnect between papers such as this one, and the way the management scholars and audience is thinking about and approaching these problems. This paper, and the broader issue it addresses, is highly relevant to those management scholars and practitioners working on supply chain decarbonization. So, engaging with this literature would help position the work more broadly and help bridge some cross-disciplinary impact.

Hope the authors find the above comments useful.

Reviewer #2

(Remarks to the Author)

All my previous comments have been appropriately addressed by the authors.

Reviewer #3

(Remarks to the Author)

The manuscript has certainly improved. However, two of my more fundamental comments that I think get to the core of the significance of the work seem to have been addressed in a superficial manner. The two comments are:

* It is well understood that corporate scope 3 estimates are highly uncertain, to the point that most companies do not even attempt to prepare "complete" estimates. A deeper issue, is that there is no objective definition of even what is a "complete" scope 3 GHG accounting. So, the distinction between national and global/regional EEIO models is one contributor to this uncertainty...a factor that is worth understanding. But, a 10% difference in values that potentially have an overall uncertainty of up to an order of magnitude larger, may not be as scientifically or policy relevant as the authors suggest. The problem is that the authors are silent on this broader issue of scope 3 estimation uncertainty and how their results fit into this very messy and high uncertainty GHG reporting landscape.

* The second point is related to trend uncertainty. The authors added some qualitative language mentioning the issue, but added no substance. The authors' two implied use cases for corporate GHG reporting, including scope 3, are to compare companies to see which one is more carbon intensive at a given time. The idea being that investors or consumers could use this information for decision-making, for example. The problem is that these corporate scope 3 reports are explicitly not designed for comparison across companies. So this is not a true use case. The other usage is to look at trends in emissions over time for each company to see if it is getting better or worse over time. For this use case then we are concerned with the uncertainty in the emissions trend estimates. For this study, then, the question is whether the estimation bias (i.e., ~10%) is a systematic bias or not. If it is systematic, then it really has little to no effect on the trend for a given company.

So I am still struggling with the use case question of this paper. A paper that just discussed the differences between national and global EEIO models would be interesting. But, when it is brought into the context of applying it for Scope 3 reporting, then one has to deal with the use cases for scope 3 and addressed how and why these differences matter.

Version 2:

Reviewer comments:

Reviewer #3

(Remarks to the Author)

Reviewer #1 (Remarks to the Author):

This study addresses a methodological problem in corporate supply chain carbon accounting. While estimating Scope 3 emissions by using company-specific primary data yields the most accurate and specific estimates, such data is typically hard to obtain and therefore is often substituted with input-output data. I-O data sources such as the US-EEIO however rely on single-regional data which, this paper argues, results in undercounting of emissions. It demonstrates this by considering the use of multiregional models as a solution using a high-resolution multiregional input-output model (CEDA). The authors quantify the discrepancies in emissions estimates for upstream supply chains of companies, finding a substantial underestimation when using a single-region (US alone) model and analyze this discrepancy across sectors.

This paper, in my opinion, adds to the evidence of the importance of using multi-regional models. To this extent, this is an appreciable contribution.

We appreciate the reviewer's careful review and constructive comments (to which we respond below).

The paper is also well written and easy to read though the explanations for Figures 2 and 3 can be more precise and need additional clarity. The analysis is competently executed, the methods clearly explained.

We have revised the text and captions describing Figures 2 and 3 in an effort to clarify the results they present.

My major concern with the paper is that while this indeed adds to the body of evidence of the importance of multi-regional models, is this sufficiently novel in and of itself? There are several studies (e.g., see Li et al. 2020 Environ. Sci. Technol. DOI: 10.1021/acs.est.9b05245) in the literature arguing for and carrying out an analysis of carbon footprinting using MRIO models – the relative contribution of the present study must therefore be carefully explained and substantiated.

My assessment of the paper will be more favorable if the authors therefore do a more thorough and careful job positioning their study with the literature to provide a more complete picture of their original contributions and arguments to the reader.

We're glad for the push to be clearer about the novelty of this work. Although the reviewer is of course correct that prior studies (including Li et al. 2020) have shown the potential for multiregional models to be used in corporate carbon accounting, the current paper underscores and quantifies the implications of *not* using them. This is important because, although it seems obvious that single-region models will give less accurate and less actionable results, the vast majority of corporate reporters are currently using such models. If multiregional modeling were widely adopted, there could be substantial changes in the priorities of corporate mitigation efforts. And this finding is especially timely because the GHG Protocol (which sets the standards by which most companies estimate their corporate footprint) is currently in the process of updating its corporate standard.

In response to the reviewer's comment, we have revised the introduction and conclusion sections of the paper to clarify its novelty and the important potential implications of our work. (We have also added a reference to the Li et al. paper).

Reviewer #2 (Remarks to the Author):

Overall comment

The manuscript NCOMMS-24-60462T entitled “Multiregional accounting of corporate carbon emissions” falls into the scope of the journal. The paper contributes to the discourse on corporate carbon emissions and the importance of using high-resolution models to accurately estimate emissions. The authors highlight the differences of corporate carbon emissions using a multiregional environmentally extended input-output (EEIO) model compared to a single-region EEIO. Applying a multiregional EEIO modelling, the authors demonstrate significant differences in emissions estimates across various sectors and regions. Their work is a valuable contribution to understanding the limitations of current corporate carbon accounting methods, particularly regarding the often underestimated impact of international supply chains on emissions footprints. To further improve the write-up and quality of the manuscript, I have provided some few comments and suggestions below:

We appreciate the reviewer’s positive feedback and helpful suggestions. We respond to each of the comments as detailed below.

1. Although the focus of the study is clear-that is, using multiregional models over single-region models in estimating corporate carbon footprints, I think the significance of the findings must be further elaborated on. The findings showing an aggregate difference of 2.0 GtCO_{2e} is indeed substantial, but further contextualization around its implications is needed. For instance, what are the potential impacts of this on corporate decision-making and regulatory compliance- this would help clarify the practical relevance of the findings.

We agree that the text could have been clearer about how our findings could be used by individual corporations and policy makers. As noted in response to Reviewer #1, multiregional resolution will often reveal different hotspots in a corporate footprint, which may in turn alter a given company’s prioritization of data-gathering and mitigation efforts. If such models were widely adopted, the aggregate changes could lead to very different and possibly more effective climate actions being undertaken.

The GHG Protocol corporate standard is also now undergoing a major revision, so that our findings could be timely evidence to support new requirements that corporations move beyond single-region EEIO models in evaluating their upstream scope 3 emissions (and reduction opportunities).

Thus, in response to the reviewer’s comment, we have revised the text to clarify how multiregional resolution could benefit specific corporations and corporate reporting in the aggregate.

2. In line 251- Authors highlight the portion of the equation representing the Leontief inverse but do not provide its meaning or definition, unlike the technology matrix where authors outline that it represents the “ratio of commodity input per unit of output”. It’s important to provide a clear explanation for non-specialist readers.

We appreciate the suggestion, and have revised the text to better describe the Leontief inverse.

3. Again, similar to my previous comment above, in line 255 authors mention emission intensity but do not provide a clear definition of what it represents.

We appreciate the suggestion, and have revised the text to clearly define emissions intensity the first time it is used, and to also include consistent parenthetical reminders when the term is used later in the text.

4. The limitations section is thorough (lines 185-206), highlighting assumptions of the EEIO data and model. However, the discussion could benefit from a more detailed exploration of why single-region models continue to dominate in corporate reporting practice, despite their known limitations. For instance, authors highlight in line 284 that of the number of companies reporting their emissions to CDP, 75% used a single-region EEIO model, and 25% a multiregional model. Identifying the barriers to adopting multiregional models and proposing potential solutions, would enhance the study's impact and practical applicability.

This is a very good suggestion, as it also helps us explain the novelty and importance of our work. We have therefore added further context of why so few corporations use multiregional models today.

We believe the main barrier has been lack of available and accessible multiregional models, in contrast to the single-region models for the U.S. and U.K. In light of this, and to further support the value and actionability of our work, we have been working to develop an open-source version of CEDA that will be freely and publicly available before this manuscript is published.

5. Although the authors detail the construction of the CEDA model and data sources, it is not clear whether they used a software for their analysis. This is particularly relevant for practitioners interested in adopting or adapting the multiregional approach presented.

We agree that it is useful to clarify how such models are implemented, and have revised the text to more thoroughly describe the tools we used, including a link to the publicly available version of the model we used.

6. The supplementary figure on selected commodity emissions intensity is valuable. However, its relationship to the main text is somewhat unclear. Briefly mentioning it within the main manuscript could provide readers with a clearer understanding of how the supplementary data supports the primary findings.

We appreciate the suggestion, and have added a sentence to the revised text that describes and references the supplementary figure (L98-100).

Reviewer #3 (Remarks to the Author):

GENERAL COMMENTS

Your manuscript is relevant to an interesting and policy-relevant issue, which is the uncertainty in and interpretation of corporate Scope 3 (category 1) emission estimates. The use of and reliance on EEIO emission factors (EFs) by companies is certainly important to understand. Unfortunately, I found your manuscript wanting with respect to both presenting a clear research question and then providing a scientific and/or policy-relevant discussion and answer.

Many of your findings seem to actually be about the differences in "spend-based" sectoral EFs between the two models you examine. This is an interesting, although I assume not a novel, observation. A discussion of how EEIO models handle (or neglect) international trade flows is a good one. Exploration of the causes

of uncertainties in these EFs would one research question to explore more, and could be done without any reference to voluntary corporate GHG inventory reporting.

Another research question would be the uncertainties and biases in voluntary corporate GHG inventory reporting that result in the use of different types of spend-based EFs and then the implications for those findings for one or more specific applications of corporate GHG reporting. It seems this is closer to what your manuscript is attempting to answer, but I did not find the discussion really explored any specific implications. The reader is then left wondering why does this matter. Is the finding that companies should shift to using a model like CEDA for all their scope 3.1 EFs? In all cases? What would be the implications of switching? You refer to how primary data is preferred but do not address the general uncertainties of using spend-based EFs from EEIO models in place of primary data.

In other words, your manuscript alludes to the findings having policy implications, but the reader is left struggling to clearly see them.

Overall, it would be useful if you would be clear and explicit regarding which specific use cases for corporate GHG reporting you are considering and referring to in your discussion. Is it just corporate net zero-type target setting and tracking? CBAM type policies? Security investor disclosure requirements/laws? Others?

We appreciate the reviewer's overall impressions and constructive suggestions, which are consistent with those of the other reviewers. As among the specific angles noted by the reviewer, we are primarily focused on how the dominance of single-region models is undermining both the accuracy and usefulness of corporate carbon accounting. Thus, we have revised the text to clarify the implications of our findings for individual corporations and policy makers, including as evidence to support the ongoing revision of the GHG Protocol's corporate accounting standards.

To further support the value and actionability of our work, we have also now developed an open-source version of CEDA that will be freely and publicly available when this manuscript is published.

We further respond to the specific comments as detailed below.

SPECIFIC COMMENTS

Abstract, L40

It would be useful to have a sense of scale for the 2 Gt value (e.g., as a percentage of CDP reported totals or something).

We appreciate this suggestion, and have added context for this number as a share of the total operational and upstream emissions reported to CDP by the companies included in our analysis (9.7%).

Abstract, L46

The conclusion seems to be focused on accuracy (reduced uncertainty), but the manuscript spends little time addressing the issue of representativeness of estimates to a particular company or the sensitivity of estimates to changes in companies supply chain (time series trend uncertainties), which one could argue is the more important dimension of accuracy for many uses of corporate reporting.

We agree that the representativeness and longitudinal uncertainties highlighted by the reviewer are important dimensions that we did not fully explore in the original manuscript. In response, we have added some new discussion of how multiregional resolution might also affect trends of estimated emissions over time, and also reflect on the importance of accurately assessing such relative changes (L185-196).

L57

I believe the GHG Protocol is now referring to scope 2 as "purchased and consumed"

We appreciate this note, and have updated our language accordingly in the revised manuscript.

L68-69

We are not aware of any company that does not rely on industry average spend-based estimates for most or all of their scope 3.1. So it is more than "often instrumental".

We agree with the reviewer that spend-based estimates are ubiquitous, but we also wanted to acknowledge the explicit preference for primary data in standards like the GHG Protocol (and mindful that some analysts are vociferously opposed to any use of such industry averages). We try to handle this by noting the "routine" use of secondary data for scope 3.1 (L35) and following up with the point that it can be "instrumental" for prioritizing primary data-gathering (L45). In response to the reviewer's comment, we've further strengthened the language in the revised text to "are instrumental" instead of "often instrumental."

L85-86

The phrasing here regarding how national inventories "need" to account for international trade effects is likely confusing or misleading. There are few consumption-based national GHG inventories and even fewer that have any policy relevance. So, it would be good to clarify that you are only referring to consumption-based inventories with this statement and that the point is irrelevant to the dominant production-based GHG inventory reporting that all countries do.

We agree that the original text was poorly worded here. In response, we've revised it to state: "*...have developed consumption-based inventories for nations and multiregional models for businesses that account for the effects of international trade*¹⁹⁻²¹"

L103-114

The discussion of the differences in estimates across sectors is interesting, however, the discussion of why seems lacking. Even if it is necessarily speculative, it would be useful to elaborate here or in the discussion section below more on the analytical or physical causes for these differences.

We appreciate the suggestion, and have expanded our discussion of sectoral differences several places in the revised text (L112-115, L122-123, L148-150).

L148-149

Again, some relative reference to the scale of this difference would be helpful.

We have added context in the revised manuscript (L164).

L166-167

This finding is interesting, but no mention of the implications for emission trends is discussed. Further, are there boundary issues between the two models? Does one model implicitly create a wider value chain boundary for an "average company" than the other model? Do these differences reflect real regional differences or just structural differences in how each model draws supply chain boundaries for a company?

As noted above, we now discuss how multiregional resolution might also affect trends of estimated emissions over time, as well as the importance of understanding such relative changes (L190-192).

The system boundaries of both the single-region and multiregional models are comprehensive, so the only difference is that the latter captures country-level differences in energy systems and economic structure. We have clarified this point in the revised manuscript (Methods)

L170-172

Few if any GHG regulations in the USA or elsewhere are administered at the company-wide level versus the installation level, especially with regard to scope 3 emissions. So, it is not clear how or why your findings would have major implications for policy/regulation.

The reviewer is of course correct that current policies and regulations governing GHG emissions are not focused at the level of companies. However, there are growing requirements that some companies report their emissions, including scope 3 (e.g., CSRD in Europe and S. California), as a proxy for climate-related financial risks. Our results suggest that reporting scope 3.1 emissions based on a single-region EEIO model would underestimate such risks and mischaracterize reduction opportunities. Regulators may therefore be interested in requiring that companies take regional differences into consideration when estimating their emissions. We have clarified these points in the revised text.

L179-181

In the case of CBAM, this is administered based on process-specific production data for the targeted major commodities. EEIO modeling is not generally used, nor are broader supply or value chain emission estimates for these commodities. So, it is not clear how or why the EEIO differences you find would be relevant to this policy.

We acknowledge the reviewer's point that CBAM itself will not assess border adjustments using an EEIO model. However, we mention it as the type of financial risk that investors could underestimate if they rely solely on emissions disclosures that fail to account for the complexities of multiregional value chains. We have elaborated on this connection in the revised text.

L198-203

The point here is an argument for the use of consequential impact methods.

We appreciate the reviewer pointing this out, and we have added a mention of the potential value of consequential approaches here in the revised text, including a reference to Brander (2022).

L207-210

This conclusion regarding scope 3 seems quite obvious for most companies that report to CDP, which is more biased towards larger multinationals. And probably for any company given international trade flows.

We agree that the importance of international differences is likely quite clear to many global companies (though perhaps not to the many less experienced companies that have only recently begun to estimate their corporate footprints). In response to the comment, we have revised the concluding text to note that this may be well-understood, and to emphasize what we think is apparently less accepted: that scope 3.1 estimates from single-region models may mislead and undermine the goals of corporate carbon accounting altogether.

Reviewer #1 (Remarks to the Author):

The authors have sufficiently addressed my comments.

I notice a few minor issues that the authors can correct or clarify.

We appreciate the reviewer's efforts and further suggestions, which we respond to below.

1. There is a reference to a forthcoming CEDA 2024 dataset with a placeholder link. I am not sure if this was intentional or a typo, please correct this.

This was intentional. The CEDA dataset was recently made publicly available here: <https://openceda.org/solutions/ceda>. We have added to the link to the revised manuscript.

2. The discussion around attributional vs consequential approaches can be clearer, the authors can cite some relevant papers that get into this distinction more.

We agree the discussion about attributional vs consequential approaches could have been clearer. We have revised the text to better discuss these approaches and include additional references [L212-220].

Fourth, companies might use multiregional resolution to seek out and switch to suppliers in less emissions-intensive regions rather than engaging to improve energy systems and processes in their existing supply chains. Although such a strategy might reduce these companies' upstream emissions footprint (as attributed to their current supply chains using existing emissions intensities), it could undermine the overall climate benefits of corporate climate action insofar as the suppliers in more emissions-intensive regions can find new corporate buyers not concerned with their GHG emissions (perhaps due to a lack of multiregional emissions accounting). This sort of leakage motivates consequential approaches that seek to evaluate the effects of an intervention relative to a counterfactual (e.g., the estimated difference in emissions when changing suppliers), which may complement attributional approaches by capturing system-wide impacts of corporate decisions (Plevin et al., 2014; Earles and Halong, 2011; Palazzo et al., 2020), but have yet to be widely adopted by the standards used for corporate emissions accounting.

3. Finally, I also suggest the authors engage more directly with the supply chain management literature on supply chain emissions and Scope 3 emissions, particularly from top management journals. In my opinion, there is a bit of a disconnect between papers such as this one, and the way the management scholars and audience is thinking about and approaching these problems. This paper, and the broader issue it addresses, is highly relevant to those management scholars and practitioners working on supply chain decarbonization. So, engaging with this literature would help position the work more broadly and help bridge some cross-disciplinary impact.

We appreciate the reviewer's suggestion, and have accordingly expanded the relevant text and included references to additional recent studies from leading management journals.

Whereas companies typically have good records of their own GHG-emitting activities and purchases of electricity (scopes 1 and 2), they often lack detailed data about the emissions related to goods and services they purchase (scope 3) —which in many cases dominate the total corporate footprint, and thus pose substantial challenges to measure and manage (Hettler and Graf-Vlachy, 2024; Vieira et al., 2024; Kaplan and Ramanna 2021).

Hope the authors find the above comments useful.

Indeed we did, and again grateful for the thoughtful feedback.

Reviewer #2 (Remarks to the Author):

All my previous comments have been appropriately addressed by the authors.

We appreciate the reviewer's helpful suggestions, which have undoubtedly improved the paper.

Reviewer #3 (Remarks to the Author):

The manuscript has certainly improved. However, two of my more fundamental comments that I think get to the core of the significance of the work seem to have been addressed in a superficial manner. The two comments are:

* It is well understood that corporate scope 3 estimates are highly uncertain, to the point that most companies do not even attempt to prepare "complete" estimates. A deeper issue, is that there is no objective definition of even what is a "complete" scope 3 GHG accounting. So, the distinction between national and global/regional EEIO models is one contributor to this uncertainty...a factor that is worth understanding. But, a 10% difference in values that potentially have an overall uncertainty of up to an order of magnitude larger, may not be as scientifically or policy relevant as the authors suggest. The problem is that the authors are silent on this broader issue of scope 3 estimation uncertainty and how their results fit into this very messy and high uncertainty GHG reporting landscape.

We apologize if our previous responses seemed superficial; we take the comments seriously and think they are helping to make the manuscript clearer and stronger. We fully agree with the reviewer that there are many reasonable criticisms of the scope 3 framework, including substantial uncertainties related to data quality and methodologies applied. But as a practical matter, several thousand major companies use the framework and publicly disclose estimates of their scope 3 emissions (e.g., nearly 4,500 companies reported scope 3.1 estimates to CDP in 2023), and many in fact use such estimates to make decisions about how to focus their climate mitigation and sustainability investments (some prominent example include Microsoft, Apple, Unilever, and PepsiCo).

We've showed that applying multi-regional models not only changes the magnitude of estimated scope 3 emissions, but in many cases alters such priorities. In addition to edits that seek to make this point even more clearly, in response to the Reviewer's comment we have also now added a worked example (presented in Supplementary Fig. S2) that underscores the potential changes in both hotspots and trends over time (further described in our response to the next comment).

* The second point is related to trend uncertainty. The authors added some qualitative language mentioning the issue, but added no substance. The authors' two implied use cases for corporate GHG reporting, including scope 3, are to compare companies to see which one is more carbon intensive at a given time. The idea being that investors or consumers could use this information for decision-making,

for example. The problem is that these corporate scope 3 reports are explicitly not designed for comparison across companies. So this is not a true use case. The other usage is to look at trends in emissions over time for each company to see if it is getting better or worse over time. For this use case then we are concerned with the uncertainty in the emissions trend estimates. For this study, then, the question is whether the estimation bias (i.e., ~10%) is a systematic bias or not. If it is systematic, then it really has little to no effect on the trend for a given company.

So I am still struggling with the use case question of this paper. A paper that just discussed the differences between national and global EEIO models would be interesting. But, when it is brought into the context of applying it for Scope 3 reporting, then one has to deal with the use cases for scope 3 and addressed how and why these differences matter

As explained above, we do think the context of Scope 3 reporting is important, as corporate emissions accounting is a very common use of EEIO models.

Panel a of Supplementary Fig. S2 shows example estimates of the Scope 3.1 emissions intensity per dollar of revenue for Indian companies in 3 different sectors over time using either a single- or multi-region model in which the emissions intensity of electricity is updated based on data in the depicted years. We confine our changes to that value alone to make perfectly clear that the changes observed are due to the model applied and nothing else (i.e. we assume no changes in the revenue, structure, and supply chains of the hypothetical companies). But because the emissions intensity of electricity in the U.S. and India have trended in opposite directions 2010-2020, these estimates do too. Regardless of the comparability across companies, then, executives and investors in these hypothetical companies would get a very different impression of their upstream carbon footprint over time.

Similarly, panel b of the figure shows that the hotspots of upstream emissions (here the top 3 contributing sectors) also differ substantially. Taken together, we hope this example helps clarify the potential effects of model choice for a given company. Uncertainties in these estimates of course remain, but companies could apply consistent methods to consistently-collected activity data and make quite different ESG decisions based on such results.

We have revised the text to include this example and point to the new figure (L213-218).

Overall comment

The manuscript NCOMMS-24-60462T entitled “**Multiregional accounting of corporate carbon emissions**” falls into the scope of the journal. The paper contributes to the discourse on corporate carbon emissions and the importance of using high-resolution models to accurately estimate emissions. The authors highlight the differences of corporate carbon emissions using a multiregional environmentally extended input-output (EEIO) model compared to a single-region EEIO. Applying a multiregional EEIO modelling, the authors demonstrate significant differences in emissions estimates across various sectors and regions. Their work is a valuable contribution to understanding the limitations of current corporate carbon accounting methods, particularly regarding the often-underestimated impact of international supply chains on emissions footprints. To further improve the write-up and quality of the manuscript, I have provided some few comments and suggestions below:

1. Although the focus of the study is clear-that is, using multiregional models over single-region models in estimating corporate carbon footprints, I think the significance of the findings must be further elaborated on. The findings showing an aggregate difference of 2.0 GtCO₂e is indeed substantial, but further contextualization around its implications is needed. For instance, what are the potential impacts of this on corporate decision-making and regulatory compliance- this would help clarify the practical relevance of the findings.
2. In line 251- Authors highlights the portion of the equation representing the Leontief inverse but do not provide its meaning or definition, unlike the technology matrix where authors outline that it represents the “ratio of commodity input per unit of output”. It’s important to provide a clear explanation for non-specialist readers.
3. Again, similar to my previous comment above, in line 255 authors mention emission intensity but do not provide a clear definition of what it represents.
4. The limitations section is thorough (lines 185-206), highlighting assumptions of the EEIO data and model. However, the discussion could benefit from a more detailed exploration of why single-region models continue to dominate in corporate reporting practice, despite their known limitations. For instance, authors highlight in line 284 that of the number of companies reporting their emissions to CDP, 75% used a single-region EEIO model, and 25% a multiregional model. Identifying the barriers to adopting multiregional models and proposing potential solutions, would enhance the study’s impact and practical applicability.

5. Although the authors detail the construction of the CEDA model and data sources, it is not clear whether they used a software for their analysis. This is particularly relevant for practitioners interested in adopting or adapting the multiregional approach presented.
6. The supplementary figure on selected commodity emissions intensity is valuable. However, its relationship to the main text is somewhat unclear. Briefly mentioning it within the main manuscript could provide readers with a clearer understanding of how the supplementary data supports the primary findings.